# Early aerial expedition photos reveal 85 years of glacier growth and stability in East Antarctica

Mads Dømgaard [1] ✉, Anders Schomacker [2], Elisabeth Isaksson[3], Romain Millan [4], Flora Huiban [1], Amaury Dehecq[4], Amanda Fleischer[1], Geir Moholdt [3], Jonas K. Andersen[1] & Anders A. Bjørk [1]

During the last few decades, several sectors in Antarctica have transitioned from glacial mass balance equilibrium to mass loss. In order to determine if recent trends exceed the scale of natural variability, long-term observations are vital. Here we explore the earliest, large-scale, aerial image archive of Antarctica to provide a unique record of 21 outlet glaciers along the coastline of East Antarctica since the 1930s. In Lützow-Holm Bay, our results reveal constant ice surface elevations since the 1930s, and indications of a weakening of local land-fast sea-ice conditions. Along the coastline of Kemp and Mac Robertson, and Ingrid Christensen Coast, we observe a long-term moderate thickening of the glaciers since 1937 and 1960 with periodic thinning and decadal variability. In all regions, the long-term changes in ice thickness correspond with the trends in snowfall since 1940. Our results demonstrate that the stability and growth in ice elevations observed in terrestrial basins over the past few decades are part of a trend spanning at least a century, and highlight the importance of understanding long-term changes when interpreting current dynamics.

The East Antarctic Ice Sheet (EAIS) contains more than 52 m of potential sea level equivalent (SLE)[1]. Recent observations indicate that the EAIS is more vulnerable than previously anticipated[2], and has made a considerable contribution to the continent-wide mass loss during the past decades[3]. The losses have primarily occurred in some of the marine-based catchments in Wilkes Land[4], and are largely attributed to the intrusion of modified Circumpolar Deep Water (CDW)[2]. The terrestrial catchments, where the majority of the ice is grounded above sea level, have recently shown a mass gain caused by increased accumulation[5–8], which has balanced some of the overall mass loss[9,10]. Observational time series of glaciers in East Antarctica pre-dating the satellite era are rare[11] and consequently not long enough to determine if recent trends are independent of natural fluctuations[2,12]. Historical datasets from early expeditions serves as a crucial link connecting

records from the pre-satellite era, such as those derived from ice cores[13] or geological[14] and geomorphological evidence[15], to quantitative observations of mass change acquired from satellites[5–7]. While geological and geomorphological records cover longer time scales with temporal uncertainties of up to thousands of years[14,15], SMB estimates from ice cores are generally very local and spatially confined[16]. In contrast, data from historical aerial expeditions often provide extensive coverage across large areas, with detailed temporal and spatial information[17–19]. Additionally, historical data provide an important baseline for forward modeling of glacier dynamics, allowing for long-term reanalysis data and more accurate model calibration[20]. In Greenland and Svalbard, long-term observations from historical aerial images have been vital for determining the historical response of glaciers to climate change[18,19,21,22]. However, in Antarctica, the scarcity

[1]Department of Geoscience and Natural Resource Management, University of Copenhagen, 1350 Copenhagen K, Denmark. [2]Department of Geosciences, UiT The Arctic University of Norway, Postboks 6050 Langnes, NO-9037 Tromsø, Norway. [3]Norwegian Polar Institute, 9296 Tromsø, Norway. [4]Univ. Grenoble Alpes, IRD, CNRS, INRAE, Grenoble INP, IGE, 38000 Grenoble, France. ✉e-mail: mld@ign.ku.dk

of historical climate data makes climate reanalysis estimates before the 1970s largely uncertain[10,23], and observed trends cannot clearly be distinguished from natural variability[24,25].

Here, we rediscover and utilize the images from the earliest large-scale aerial photography campaign conducted on the Antarctic continent, allowing us to extend the era of observational records of glacier evolution back to the 1930s. Since the beginning of the 20th century, several expeditions were launched to Antarctica with the aim of exploring and capturing aerial images for the production of geographical maps[26–30]. However, just a handful of studies have previously used these data for generating digital elevation models (DEMs) and only for glaciers located in West Antarctica and the Antarctic Peninsula[11,31,32], dating back to 1947[32]. On the Antarctic Peninsula, these observations show widespread near-frontal surface lowering and inland stability since 1960[31]. On the other hand, historical observations of the Byrd Glacier over the past 40 years indicate a constant surface elevation, stable grounding line, and surface flow velocity[11].

Here, we study 21 glaciers located in three regions along c. 2000 km of the EAIS, from Lützow-Holm Bay (38° East) to Ingrid Christensen Coast (79° East) (Fig. 1A). All glaciers are marine-terminating outlets of the EAIS, varying in width from 2 to 10 km, and with the fastest flow speeds reaching 2 km/yr. Some have large floating ice tongues, while others have their frontal position close to the grounding line. The glaciers are located in basins containing around 2.6 M km³ of ice (SLE 7.23 m), and the specific sub-regions studied (Fig. 1A) contain 0.42 M km³ of ice (SLE 1.15 m)[1]. In recent decades, these regions have maintained balance or gained mass[3], leading to advances in terminus[33] and grounding line position[34,35], during a period of substantial mass loss and terminus retreats occurring at other sectors of the Antarctic Ice Sheet[3,5–7,33,34,36–39].

In late 1936, the Norwegian whaling entrepreneur Lars Christensen initiated his fifth and final expedition (Thorshavn IV) to Antarctica with the specific aim of capturing aerial images for producing the earliest detailed maps of the East Antarctic coastline. Drawing inspiration by the Greenland aerial expeditions in 1932[40] and the aerial mapping of Svalbard in 1936[19], the coastline was photographed with an oblique angle and a stereo overlap of c. 60%. The expedition acquired 2200 photographs covering approx. 2000 km of the coastline from 82° to 20° East[30] (Fig. 1A and Supplementary). Twelve topographic maps of the coastline were produced from the images. However, they were not published until 1946 due to the German occupation of Norway. Since then, the images have been stored at the Norwegian Polar Institute in Tromsø and largely forgotten[41,42]. Luckily, the film has been kept at optimal conditions, giving us the best possible starting point for recreating the historical East Antarctic glacier conditions with modern digital technologies.

The images from the Norwegian 1936-37 expedition are unique, as they provide the earliest detailed view of a regional Antarctic coastline and allow historical reconstruction of glaciers in East Antarctica. The image archive poses considerable challenges for analysis due to its limited camera metadata, sparse flight line and camera position information. Consequently, a comprehensive manual image selection and geolocation process is essential prior to conducting any analysis (Methods). Moreover, the vast majority of the images are captured from low elevations or large distances from the coast, have poor image contrast and/or are overexposed. Additionally, the majority of East Antarctic coastline lacks visible bedrock features, which are essential for producing reliable results (Methods). Nevertheless, we have successfully identified 128 images, with sufficient overlap, contrast and bedrock presence for ground control. The Norwegian 1936/37 imagery is combined with more recent historical aerial photographs (1956–1973) from Australian aerial campaigns (Fig. 1) and also with modern satellite data. This allows us to quantify changes on a decadal timescale throughout the 20th and 21st Century.

We utilize a total of approximately 300 aerial images to examine changes in glacier elevations, velocities, and terminus positions (Methods). We reconstruct past ice sheet configurations by processing the historical aerial images using structure-from-motion (SfM) photogrammetric techniques[43,44], to provide the, to date, most extensive historical assessment of regional glacier dynamics in Antarctica. The SfM models are georeferenced to real-world coordinates by GCPs across the study area and transferring the coordinates from modern high-resolution stereo satellite imagery[45]. The uncertainty of our

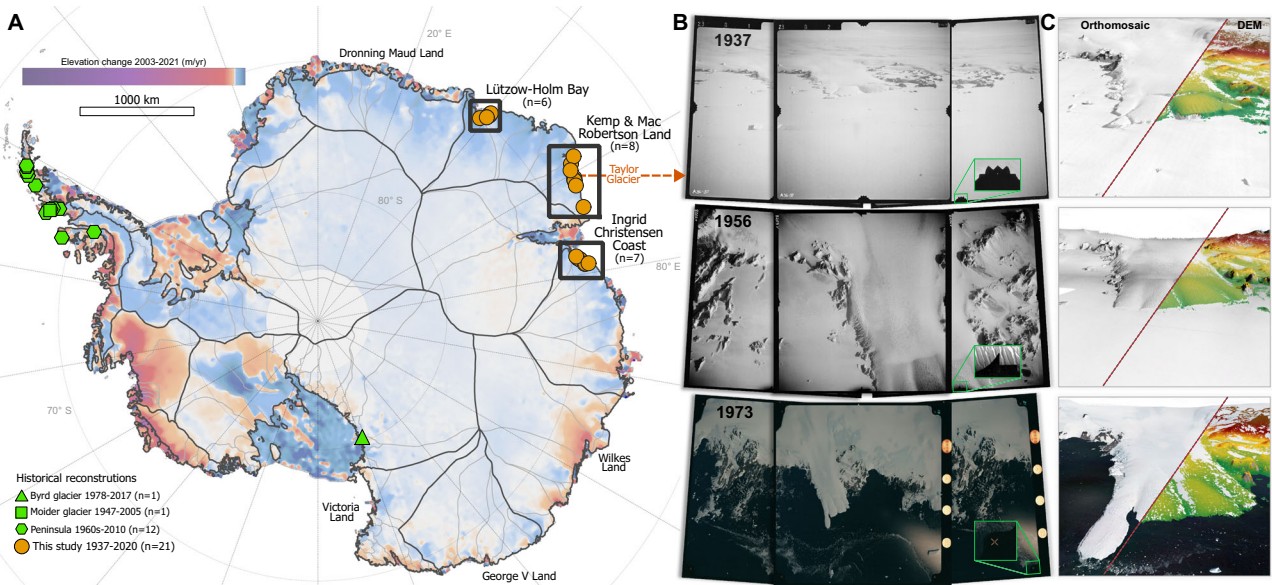

**Fig. 1 | Historical glacier reconstructions and aerial images. A** Existing historical glacier reconstructions (Byrd[11], Moider[32] and Peninsula[31]) and glaciers included in this study (frontal reconstructions n = 21, elevation reconstructions n = 12, velocity reconstructions n = 4), overlaid on 2003–2021 Antarctic annual elevation change from Smith et al[5]., with MEaSUREs basin[72]. **B** Taylor glacier in 1937, 1956, and 1973 as captured in the aerial images. Close up shows the different types of fiducial marks used for standardizing the internal image geometry. **C** Produced digital elevation models (DEM) overlain on orthomosaics generated from interpolated DEMs. For the productions of the 1956 DEM and orthomosaic we included additional oblique images (not included here) as the glacier was photographed with a trimetrogon camera setup.

historical reconstructions is determined by comparing the historical surface to a modern reference surface over stable bedrock (Methods).

From the aerial imagery we are able to quantify elevation changes of 12 glaciers (Methods). The produced DEMs vary in pixel size from 8 m to 25 m, and the elevation uncertainty ranges from 1.6 m to 9.9 m depending on the imagery and location (Methods). Based on the historical DEMs, we perform orthorectification of the aerial images and generate orthophoto mosaics with a pixel size ranging from 1 to 8 m. Due to the extensive collection of aerial images captured in the 1950s, we are able to measure the historical flow speeds of four glaciers, where the image reacquisition interval successfully matches glacier flow speeds. Lastly, we determine frontal changes of 21 glaciers from 1937 to 2023 (Table S1 and Fig. S11). From the 85 years of observations, we find two distinct regional patterns; one of constant glacier surface elevations and one of ice thickening.

## Results

### Historical frontal retreat and constant long-term ice elevations in Lützow-Holm Bay

Despite large variations in size and characteristics, all six glaciers in Lützow-Holm Bay experienced a net retreat between 1937 and the 1980s, when they reached an almost simultaneous minimum (Fig. 2) associated with a complete break-up of fast ice in the bay[35,46]. Another smaller retreat phase occurred around the mid-2000s, with the exception of Shirase, and again during 2016-18[47]. By far, the largest fluctuations are observed at Shirase Glacier with a total range of almost 90 km in the ice-front position from its 1963 maximum to its minimum in 1988. In contrast, Langhovde and Hovdebreen Glacier show frontal variations of less than 1 km between 1937 and 2023. Since the retreat in the 1980s, all glaciers except Langhovde and Honnörbrygga have, at some point, advanced to a similar marginal position or even extended beyond their extent recorded in 1937. The frontal collapse of Honnörbrygga Glacier from 1937 to 1956 was observed in 1964[42], but has never before been quantified. From 1956 to 1974, Honnörbrygga Glacier retreated an additional 4.5 km, resulting in a total retreat of more than 10 km. Since 1974, the glacier has been able to regrow less than half of its total retreat. From its current position, it would require 17 years of uninterrupted advance at its current flow speed in order to reach its 1937 maximum extent.

The retreat of Honnörbrygga Glacier has not led to a thinning of its grounded section between 1937 and 2020. The mean surface elevation change rate during this period is insignificant at +0.01 m/yr ± 0.10 m/yr from 1937 to 2016 and +0.24 m/yr ± 1.08 m from 2016 to 2020. A similar pattern of unchanged elevations is evident all along the Syowa Coast from 1937 to 2016 (Fig. S12A), with rates of +0.01 m/yr ± 0.10 m/yr and +0.03 m/yr ± 0.10 m/yr at Langhovde and Hovdebreen Glacier, respectively (Fig. 2). Between 2016 and 2020, DEM differences indicate a slight negative rate of change, −0.43 m/yr ± 0.74 m/yr for Langhovde and −0.25 m/yr ± 0.74 m for Hovdebreen. Consequently, the absolute elevations have remained unchanged throughout the entire observation period from 1937 to 2020 (Fig. S23). Modern altimetry-based observations covering the years 2003 to 2021[5] confirm limited surface elevation changes along the Syowa Coast, with rates ranging from +0.02 m/yr ± 0.01 m/yr to +0.06 m/yr ± 0.05 m/yr, except for Shirase Glacier, which exhibited thickening at a rate of +0.53 m/yr ± 0.27 m/yr.

### Regional frontal fluctuations and long-term ice thickening

Our observations show no regional long-term trend in the frontal positions of the studied glaciers in Kemp Land, Mac Robertson Land, and along Ingrid Christensen Coast between 1937 and 2022. The glaciers fluctuate between periods of frontal advances and retreats of varying distances (<0.1 km to 13.5 km) and intervals (3−50 years). Most noticeable is the 13.5 km advance of Mulebreen Glacier since the 1980s and the retreat of Jelbart Glacier, which is 3.5 km short of its 1937 marginal position.

From the historical DEMs, we observe an overall thickening of the glaciers in Kemp and Mac Robertson Land between 1937 and 2021 and along Ingrid Christensen Coast between 1960 and 2021 (Fig. 2, Fig. S23). In Kemp and Mac Robertson Land, the largest changes are observed at Hoseason and Taylor Glacier, with an annual average surface elevation increase of +0.23 m/yr ± 0.07 m/yr and of +0.11 m/yr ± 0.05 m/yr since 1937, respectively. In comparison, Utstikkar, Jelbart, and Brunvoll Glacier show less thickening since 1937, with rates of +0.06 m/yr ± 0.04 m/yr, +0.04 m/yr ± 0.04 m/yr, and +0.04 m/yr ± 0.22 m/yr, respectively (Fig. S23). For Jelbart and Utstikkar Glacier, the thickening is substantially larger ( + 0.11-0.14 m/yr ± 0.04 m/yr since 1937 and +0.13-0.42 m/yr ± 0.04 since 1973) when excluding observations from the hummocky dynamic frontal regions where surface undulations result in short-term dH variations of up to ±20 m. (Fig. S15).

Along Ingrid Christensen Coast, the largest thickening is observed at Shennong Glacier with an average rate of +0.17 m/yr ± 0.06 m/yr from 1960 to 2011 (Fig. 2). At Flatnes, Hovde, and Brown Glacier, the rates are smaller, ranging from +0.06 m/yr ± 0.02 m/yr to +0.11 m/yr ± 0.05 m/yr (Fig. 2). Between 2011 and 2021, DEM differences show negative rates for Hovde and Flatnes Glacier (−0.23 m/yr ± 0.15 m/yr and −0.07 m/yr ± 0.15 m/yr), positive rates for Shennong (+0.26 m/yr ± 0.15 m/yr), and rate close to zero at Brown Glacier. In total, all glacier along Ingrid Christensen Coast experienced an absolute elevation increase from 1960 to 2021/2022 (Fig. S23). Despite the long-term thickening of the glaciers in Kemp and Mac Robertson Land and along Ingrid Christensen Coast, our results reveal variations in the magnitude of the rates and periods of thinning. Specifically, Taylor and Utstikkar Glacier experienced thinning between 1956/1960 and 1973, Jelbart from 1937 to 1973, and Hoseason, Flatnes, and Hovde exhibited thinning during the past decade (Fig. 2). These findings suggest low magnitude decadal variability superimposed on a pattern of long-term thickening.

Our analysis of historical velocities of glaciers in Kemp and Mac Robertson Land reveal a pattern of constant flow velocities since the 1950s (Fig. 3). The historical velocities have an uncertainty range of 7.2 m/yr to 10 m/yr, and for Taylor, Jelbart, Hoseason and Utstikkar Glacier, the historical velocities fall within the range of modern satellite-derived velocities from 2006-2018, even when disregarding the 2006-2013 annual velocities with a high level of uncertainty.

## Discussion

The absence of pronounced regional trends in frontal position in Kemp and Mac Robertson Land and along Ingrid Christensen Coast is in line with existing observations of basin-wide median frontal movement rates between 1974 and 2012[33]. Historical observations of glacier terminus positions from other regions in East Antarctica reveal cyclic behavior with no overall trend from the 1950s to the late 1990s[48]. Contrary, on the Antarctic Peninsula, the majority of glaciers have retreated since the 1950s with a suggested link to atmospheric warming[39]. In Wilkes Land, East Antarctica, a recent anomalous frontal retreat has been linked to a reduction in sea ice[33]. Additionally, landfast sea ice played an important role in the observed simultaneous frontal retreat of the glaciers in Lützow-Holm Bay in the 1980s[35,46] and again from 2016 to 2018[46,47]. Moreover, the 2016-2018 retreat was coupled to a decrease in surface elevation and an acceleration in flow at Shirase, Skallen, and Telen Glacier, whereas Honnörbrygga and Langhovde Glacier were unaffected[47]. Similarly, our findings of a long-term frontal retreat at Honnorbrygga and Langhovde Glacier since 1937 does not coincide with any changes in the surface elevation of these glaciers (Fig. 2), suggesting that these floating ice tongues have provided limited buttressing on a decadal time-scale. Additionally, our results indicate that the land-fast sea-ice conditions controlling the frontal position of Langhovde and Honnörbrygga Glacier have become more susceptible to break up during the past 85 years. Notably, recent findings have pointed towards a possible link between these localized

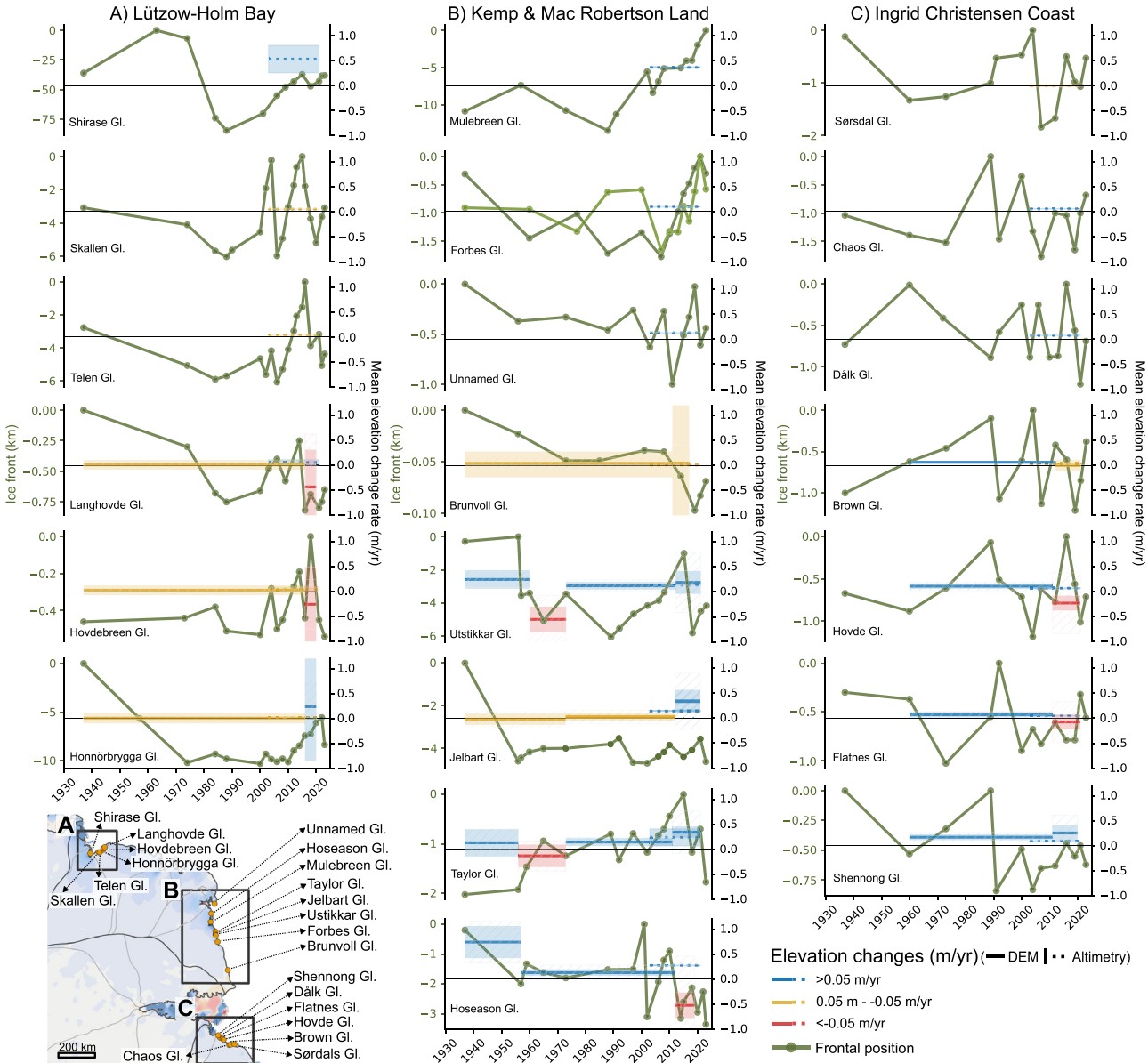

**Fig. 2 | Changes in ice-front position and elevation. A** Lützow-Holm Bay, (**B**) Kemp & Mac Robertson Land, and (**C**) Ingrid Christensen Coast. The changes in frontal position is shown relative to the glacier-specific maximum. Forbes Glacier is shown with two frontal positions, as it has two separate floating extensions. Changes in elevation are shown as the annual rate of change in m/yr between successive observational periods, with the shaded area indicating the model uncertainty (1σ) of the produced digital elevation models (DEM) (Methods). The hatched areas represent the sampling uncertainty (2σ + 1.5 m) associated with the sensitivity of the calculated changes (Supplementary). Note that the y-axis values for frontal positions differ among glaciers.

sea-ice breakups and the detection of Warm Deep Water (WDW) beneath Langhovde Glacier in 2018[49].

The terrestrial regions of the EAIS respond mainly to atmospheric forcing[2]. Overall, there has been no significant trends in annual or seasonal mean air temperature in East Antarctica since the 1950s[50], and mean austral summer air temperature (December to February) from stations in all regions rarely exceeds 0 °C (Fig. 4C). This suggests that surface melting have played a minimal role in the documented ice thickness changes overtime. However, at Davis station we do observe periods of above zero-degree temperatures during the 1970s and 2000s, which have caused intervals of increased ablation in this region near sea level.

While previous research on Antarctic snowfall found no statistically significant changes since the 1950s[51], recent studies utilizing compiled data on ice core records indicate increased Antarctic-wide snow accumulation during the past 200 years[10], with links to atmospheric warming[52], ozone depletion[53], and a positive shift in Southern Annular Mode (SAM)[10]. Nevertheless, there are notable regional differences in accumulation in both sign and magnitude, and in East Antarctica the results are derived from only a few ice core records[10].

ERA5 reanalysis data suggest a consistent positive long-term trend in mean annual snowfall along the coastline of Kemp and Mac Robertson Land, and Ingrid Christensen Coast since the 1940s, whereas in Lützow-Holm Bay, snowfall has remained almost constant (Fig. 4B). The long-term trend is most pronounced in Kemp and Mac Robertson Land, where snowfall increased by ~50 % between 1940 and 2022, corresponding to 17.3 mm of water equivalent (w.e.) per decade, whereas along Ingrid Christensen Coast, the increase is around 15% equivalent to 5.3 mm w.e. per decade. Thus the trend in snowfall corresponds with the observed long-term historical changes in glacier elevations in each of the respective regions (Fig. 4). Observation-based

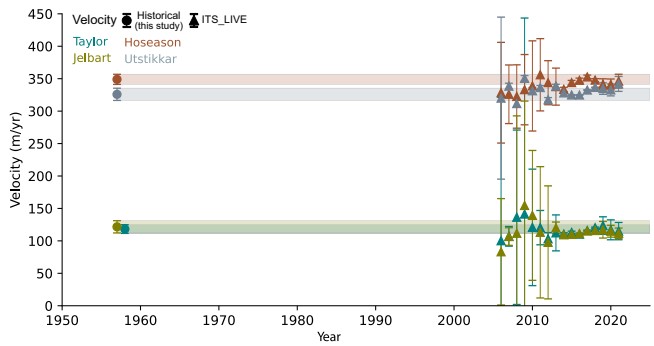

**Fig. 3 | Historical and modern velocities of Taylor, Hoseason, Jelbart, and Utstikkar glaciers.** The shaded area shows the uncertainty in historical velocity estimates, defined as the combined the uncertainties of the mean square positional error (MSPE) and the manual error associated with feature positioning (Methods). Historical velocities are estimated by manually tracking the movement of crevasses between sets of orthophoto mosaics. Modern velocities and associated uncertainties are extracted from the ITS_LIVE annual velocity mosaics[67] (Methods).

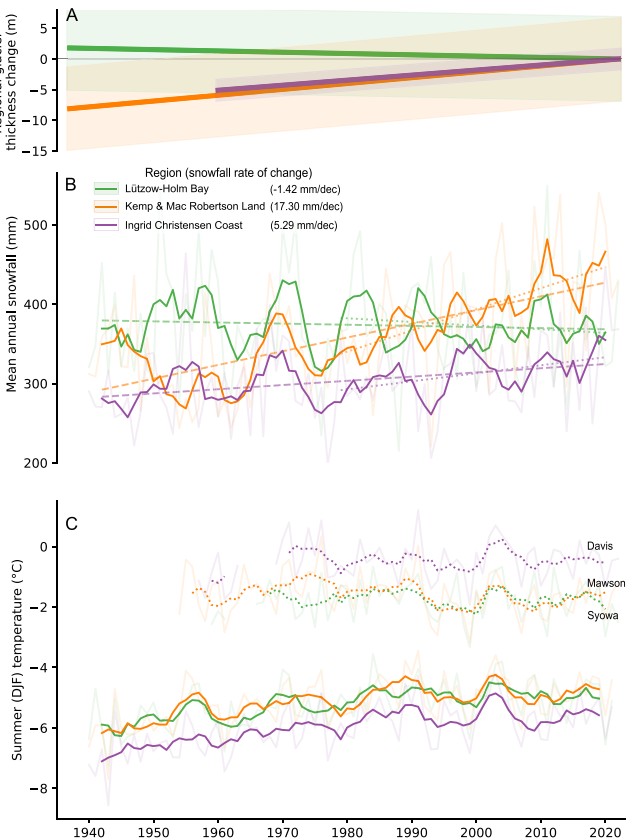

**Fig. 4 | Regional glacier elevation changes, mean summer temperature and annual snowfall. A** Glacier elevation changes are calculated as the mean elevation change (dH) over the longest period for each region. **B** ERA5 mean annual snowfall in mm water equivalent (w.e.) from 1940 to 2022 for each region (Fig. S21A), plotted as 5-year running average. Linear trend is calculated based on the running average since 1940 (dashed line) and 1979 (dotted line). Rate of change is given for the period 1940–2022 in mm per decade. **C** Mean summer air temperature (December, January, and February) from station measurements (dotted line) and ERA5 2 m temperature data, plotted as 5-year running average.

atmospheric reanalysis data of Antarctica are questionable prior to the satellite era due to a lack of historical data[23], particularly concerning snowfall[10,51]. The trend in snowfall since 1979 is nearly identical trends to that from 1940 for all regions (Fig. 4). However, to differentiate the increase in snowfall from long-term background noise, larger trends or longer-time series are needed[25]. Nevertheless, we hypothesize that the observed increase in surface elevation since 1937 in Kemp and Mac Robertson Land (Fig. 4) is likely a result of changes in precipitation patterns. The absence of dynamic ice changes, observed from stable frontal positions since the 1930s and flow velocities since the 1950s (Figs. 2, 3) suggest that the observed ice surface increase results from the increase in precipitation (Fig. 4). The periodic thinning and decadal variability observed within the long-term trends in glacier surface elevation in all regions (Fig. 2), demonstrates the natural variability inherent in these systems. Moreover, it highlights the importance of long-term glacier records, as studying these short-term changes without the long-term perspective could lead to erroneous conclusions.

Basin-wide altimetry-derived elevation changes of each of the studied sub-regions show an overall increase in elevation between 1985 and 2022[7] (Fig. S22). This trend is consistent with the observed historical glacier thickening in Kemp and Mac Robertson Land and along Ingrid Christensen Coast. Contrary, in the Prince Olav Coast sub-region, which includes the Lützow-Holm Bay area, the basin-wide trend deviates from the observed long-term constant ice elevations. This suggests that the basin-wide increase is not significant on a long-term scale, or more likely, that the constant ice elevations in Lützow-Holm Bay are largely confined to this region. Other parts of Dronning Maud Land have experienced a substantial mass gain in the past two decades caused, in part, by anomalously high snowfall in 2009 and 2011[5,6,8]. Additionally, the recent mass gain of the Shirase Glacier (Fig. 2) has been linked to strengthening easterly winds reducing the inflow of modified Circumpolar Deep Water and decreasing basal melt rates[35].

Our historical observations of ice-thickness changes provide valuable insights into historical mass balance estimates of East Antarctica, as in situ records of mass balance are extremely few in Antarctica. Currently, the earliest ice-sheet wide mass balance estimates start in the late 1970s[3,6,7], and since then all the sub-regions examined in this study have exhibited either an overall mass gain or been relative unchanged. Given that our historical reconstructions extends beyond the era of reliable climate reconstruction, and considering the limited magnitude of the observed long-term changes and their localized spatial resolution, we are unable to further deduct the specific drivers of the observed changes. Regardless of potential climatic changes, our

results indicate that the glacier in Kemp and Mac Robertson Land and along Ingrid Christensen Coast, have accumulated mass during the past 85 years which inevitably have mitigated parts of the more recent mass loss from the marine basins in East Antarctica and the West Antarctic Ice Sheet (WAIS). This positive accumulation trend and positive mass balance is anticipated to persist as snowfall is expected to increase over the entire EAIS in the next century[54,55], and ice sheet modeling studies project positive mass balance estimates in all three sub-regions across all future RCP scenarios[56].

## Methods

### The 1936-37 aerial photographs

The original negatives from the 1936-37 Thorshavn IV expedition were digitized using a photogrammetric scanner at a resolution of 0.014 mm. Aerial photographs captured with a similar camera type have successfully been used for reconstructions of local and large-scale terrain models in Svalbard[19,57]. However, the Antarctic imagery contains limited camera calibration information and none or very rough camera positions. Consequently, before initiating the image processing phase, we carefully examined each image from the expedition and manually tried to determine the geolocation of where along the coastline the image was captured. However, due to the absence of discernible surface features such as bedrock, ice rises, or distinctive coastline features along large parts of the East Antarctic coastline,

**Table 1 | Overview of all utilized expedition images**

| Expedition | Year | Camera | Image type | Images used |
|---|---|---|---|---|
| Thorshavn IV | 1936–1937 | Zeiss | Oblique | 128 |
| ANARE | 1954–1965 | K17 or Eagle V | Trimetrogon | 155 |
| National mapping | 1970s | Wild RC9 | Vertical | 9 |

geolocation was not always possible. Additionally, a large quantity of images are (i) captured from low elevations or considerable distances from the coastline, or (ii) exhibit poor contrast or overexposure, which further challenged the process (Fig. S28). Consequently, our selection process led us to identify 128 images (Table 1) from the archive that aligned with the quality standards required for historical reconstructions and frontal mapping.

### The 1954–1973 Australian image campaigns

The Australian Antarctic Data Centre holds a vast collection of aerial photographs acquired since the 1950s covering the Australian Antarctic Territory. The first major period of Australian aerial photography acquisition, from 1954 to 1965, was carried out by the Australian National Antarctic Research Expedition (ANARE) using Auster and Beaver aircraft and a K17 trimetrogon camera. From 1960, the vertical K17 in the trimetrogon system was replaced with a Wild RC9. Descriptive studies were published more than 50 years ago using the trimetrogon imagery[41,58,59]. In the 1970s, the Division of National Mapping carried out extensive photographing of the areas around Prince Charles Mountains and Enderby Land using a Wild RC 9 camera from a Pilatus Porter aircraft. The archive consists of digitized negatives scanned using a photogrammetry-grade scanner, and all images contain sparse camera calibration information and imprecise camera locations.

### Structure for motion (SfM) processing

The aerial images are processed in Agisoft Metashape v. 1.7.4 using a standard SfM workflow[19,44]. We primarily process photographs from each period and glacier separately. However, in cases where glaciers are located in close proximity (such as Langhovde, Hovdebreen and Honnörbrygga, as well as Jelbart and Utstikkar), we combine the images into a single model for each period.

As a first step, we standardize the internal geometry of each image set by detecting the image fiducials and determining their mean position. Agisoft Metashapes Detect Fiducial function automatically identifies the fiducial center pixels while at the same time masking out the image frame[60]. The images are automatically scaled and oriented to ensure alignment of the fiducials. We manually inspected all fiducial positions and corrected any misplaced ones. All fiducials from the ANARE archive were manually placed. Next, we align the images by extracting up to 50,000 key points per images at Medium accuracy setting, meaning that photos are downscaled by a factor of four. For some image sets, additional tie points are manually placed to ensure the correct alignment of the images. The tie points are distinctive surface features (e.g., crevasses) that could be detected in multiple images. The image alignment solves for the unknown camera parameters, including the focal length, while estimating relative camera locations and orientations[44].

For absolute georeferencing of the 3D model, GCPs are extracted from the Reference Elevation Model of Antarctica (REMA)[45,61] and manually placed in the aerial images. The GCPs are distinct features of stable bedrock with specified (x,y,z) locations. When possible, we use the same GCPs for glaciers photographed multiple times. As the amount and distribution of stable bedrock for GCP selection varies greatly among all of our study sites, an ideal distribution is not always possible[11,44]. Following GCP placement, a full bundle adjustment is executed on the aligned block which refines the exterior and interior camera orientation parameters as well as the tie point coordinates[44,60].

We perform a dense multi-view stereo (MVS) reconstruction, with a High quality setting, meaning that dense cloud is generated at 1:2 of the original image resolution, due to processing time. The point filtering is set to Moderate to Aggressive setting for outlier removal[19]. The dense cloud is imported into CloudCompare (v. 2.12) for additional filtering, removing points outside of the local minimum and maximum as well as obvious erroneous elevations. The point clouds are gridded into DEMs with a spatial resolution ranging from 6 to 25 m, reflecting the ground sampling distance of the point cloud. Lastly, orthophoto mosaics are produced with a resolution of 2–10 m. The lowest resolution is found in products derived from the 1936/37 photographs. In cases where the image quality is poor or bedrock visibility is limited, we are only able to generate reliable orthophoto mosaics and not DEMs.

### DEM post-processing

All produced DEMs are co-registered to the REMA reference DEM using the method developed by Nuth and Kääb[62]. Following co-registration, we assess the absolute model accuracy by calculating the Normalized Median Absolute Deviation (NMAD) of the elevation differences over stable bedrock[63,64]. The accuracy vary from 1.6 to 9.9 m, with the 1936/37 DEMs being the least accurate. From the residual elevation difference values we compute spatial variograms for each of our produced DEMs to assess the degree of spatial autocorrelation in the models[19,64]. Using the xDEM python package[65], we sample empirical variograms and fit a double-range variogram model consisting of a short-range Gaussian model and a spherical model at long ranges. Finally, we propagate the pixelwise uncertainties to the estimation of the overall uncertainty of the mean elevation change of each glacier[19,64].

### Estimation of glacier elevation changes

Due to a combination of insufficient image overlap, oblique view-angle, and low-contrast snow areas, the DEMs contain areas of no data (e.g., Fig. S12–S19). To account for data gaps and enable comparison of ice elevations across multiple periods, we calculate elevation changes (dH) of the entire glacier area as well as within 150-m radius circles, all located within the minimum boundary area defined by the GCPs (Fig. S12–S19). These circles are manually positioned across the glacier area in locations with a high density of elevation observations in all periods and with an aim of capturing the overall characteristics of the glacier (Supplementary). As a result, the number of circles placed varies between the glaciers. All elevation changes are calculated relative to the REMA reference DEM strip, as differencing two historical DEMs with data gaps results in more areas with no data. We incorporate a conservative uncertainty estimate of 1 m for the REMA DEM[45] into our calculations. By comparing the historical DEMs to the reference strip, we can ensure a more comprehensive assessment of elevation changes across the study area. Subsequently, we calculate the elevation changes between each historical period as:

$$dH(period2 - period1) = dH(period2 - period3) - dH(period1 - period3)$$

(1)

where period3 is the REMA reference DEM strip. Finally, to determine the mean elevation change for each glacier, we compute the average dH of all circles placed within the glacier area, rather than using a hypsometric-weighted average, as the sampled variance in glacier surface elevation is limited and without a significant correlation to the observed dH (Fig S24). This approach allows us to extract valuable information from the sparse DEMs and assess changes in glacier elevation over time. Post-2010 elevation changes derived from DEMs are calculated by subtracting co-registered REMA DEM strips. We test the sensitivity of our results to the placement of the circles by calculating the mean dH with a subsample of the circles, as well as by slightly shifting the positions of the circles (Supplementary).

Additionally, we extract ICESat and ICESat-2 derived elevation changes between 2003 and 2021 from Smith et al.[5]. We determine the elevation change of each glacier as the mean of the pixels closest to the grounding line.

### Frontal position

We manually digitize the historical glacier frontal positions (1937–1973) from the produced orthophoto mosaics as well as from a combination of georeferenced aerial images and historical maps. Frontal positions from 1974 to 2022 are mapped from Aster and Landsat satellite images (Fig. S25, S26, and S27). In cases where a glacier is heavily fractured, we define the frontal position as the crevasse that decouples the inner section from the outer section of the floating extension. We measured the frontal changes along an approximate centerline and relative to the maximum position of the terminus[66].

### Velocity

By manually tracking the movement of glacial surface features between sets of orthophoto mosaics, we estimate historical ice velocities for Hoseason, Taylor, Jelbart, and Utstikkar Glacier in the late 1950s. These features are distinct crevasses that are easily identifiable in both sets of orthophoto mosaics, and are mainly found at the frontal part of the glacier. The number of detected features varies between glaciers, ranging from 4 at Taylor Glacier to 13 at Utstikkar Glacier (Fig. S20). For each glacier, we calculate the mean displacement of all sets of corresponding features and determine the yearly velocity using the time between image acquisitions, ranging from 4 months at Hoseason to 4 years at Taylor. The errors from the manual tracking are determined by combining the uncertainties of the mean square positional error (MSPE) and the manual error associated with feature positioning. The manual error is set to 1 pixel to account for the limitation of placing features at subpixel levels. The MSPE is based on the x- and y-errors of the estimated GCP locations for each of the orthophoto mosaics, as reported in Agisoft Metashape. For 2006-2018 we use ITS_LIVE annual velocity mosaics[67,68] and calculate the mean velocity and error of all pixels found within the glacier surface area used for the historical velocity estimates (Fig. S20S).

### Climate data and surface mass balance (SMB)

For each region of interest (Fig. S21A), we calculate the mean austral summer temperature (December to February) and mean annual snowfall between 1940 and 2022 from 0.25° ERA5 data[69,70]. Additionally, we calculate the mean austral summer temperature based on observational records from Syowa, Mawson, and Davis Station from the Scientific Committee on Antarctic Research (SCAR) Met READER project. RACMO2.3p2 annual surface mass balance (SMB) estimates[71] are extracted from 1979 to 2022 using the same regions of interest as for the ERA5 data.

## Data availability

The produced DEMs, orthophoto mosaics, ice front positions, surface flow velocity, and dH estimates are accessible via figshare (https://doi.org/10.6084/m9.figshare.23552079.v1). The REMA DEM can be downloaded from https://www.pgc.umn.edu/data/rema/. Monthly mean surface air temperature records from Syowa, Mawson, and Davis Station are available via https://legacy.bas.ac.uk/met/READER/. In addition, mean monthly ERA5 data on 2-m air temperature and snowfall are available from https://cds.climate.copernicus.eu/cdsapp#!/dataset/reanalysis-era5-single-levels-monthly-means. RACMO data is available upon request to Michiel R. van den Broeke and Melchior van Wessem. The Norwegian and Australian aerial images are available upon request to The Norwegian Polar Institute and The Australian Antarctic Division, respectively, owing to their ownership of the image rights.

## Code availability

Codes used to produce the figures of this paper are accessible at (https://doi.org/10.6084/m9.figshare.23552079.v1).

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

## Acknowledgements
This work was funded by the Villum Foundation (Villum Young Investigator grant no. 29456). M.D. acknowledges support from Augustin Fonden and Julie von Müllens Fond. We thank the Australian Antarctic Division for the supplying the Australian historical images, Michiel R. van den Broeke and Melchior van Wessem for providing RACMO SMB, and The Polar Geospatial Center (PGC) at the University of Minnesota for providing the REMA DEM.

## Author contributions
M.D., A.S., and A.A.B., designed the study. M.D. led the data analysis, with inputs from A.D., A.F., J.K.A., R.M., and A.A.B. All authors (M.D., A.S. E.I., R.M., F.H., A.D., A.F., G.M., J.K.A., and A.A.B.) contributed to writing the manuscript and interpreting the results.

## Competing interests
The authors declare no competing interests.
