## [Peer Review File · Nature Communications]

Early aerial expedition photos reveal 85 years of glacier growth and stability in East AntarcticaREVIEWER COMMENTS

Reviewer #1 (Remarks to the Author):

In this manuscript, Domgaard et al. analyze a set of historical aerial photographs (dating back to 1936) of East Antarctica and apply structure-from-motion and image correlation techniques in order to quantify historical elevation change and glacier surface velocities. In the discussion, Domgaard et al. compare their reconstructions of surface elevation change to weather reanalysis data from East Antarctica to suggest that the contrasting positive vs. negative mass balance behavior of their investigated sub-regions can be explained by regional trends in snowfall. I found this manuscript to be clear and well-written. I reviewed two previous iterations of this paper, and I found this most recent version to be the most thorough, thoughtful, and well-balanced (as much as one can expect for a short-format paper!). I have no remaining major reservations and recommend publication. See below for minor comments and line edits.

General comments:

I found the description of the data and methodology to be significantly improved from previous versions! I also found the format of a "Data and Methods" section at the end of the main text to be well-suited to this paper. This format allows for the main text to be relatively streamlined (not bogged down by the methodology), but also prevents relegating the methods to the Supplementary Information, which I think would do a disservice to the research because the curation and analysis of these historical images is the primary contribution of the paper.

There may be an error in lines 320-321, where the authors say "...we align the images...at Medium accuracy setting, meaning that photos are downscaled to half of its original resolution". At least in the older versions of Agisoft Photoscan that I'm familiar with, the setting "Very high" (or "Ultra") corresponds to no downscaling, "High" corresponds to downscaling by a factor of 2, and "Medium" corresponds to downscaling by a factor of 4. That's a factor of 4 in each dimension, so in total the image will be 1/16th of its original size. Is it the same for the latest model of Agisoft Metashape? That could be something to double-check before publishing...

Line comments (minor):

Line 13: I might suggest a slight modification of your current phrasing: "In order to determine if recent trends are independent of natural variability..." Perhaps it is more precise to say: "In order to determine if recent trends exceed the scale of natural variability"?

Line 21: add "in this region" or "in this sector". So, the full sentence will read: "The aerial images reveal that the current absence of mass loss in this sector can be extended back 85 years."

Lines 25-26: It might be easier to read if you split this phrase ("...and link observed ice thickness changes to regional long-term trends in snowfall since 1940") into its own sentence. Perhaps: "At Kemp and Mac Robertson and Ingrid Christensen Coast, we link observed ice thickness changes to regional long-term trends in snowfall since 1940."

Line 43: change "through" to "from"

Line 50: replace "historical climate data" with "historical climatic and glaciological data"

Line 53: "exploit" is an aggressive term. Replace with "leverage"?

Line 57: The word "data" should be plural: replace "this data" with "these data" or "these images"

Line 64: Modify (Fig. 1) to (Fig. 1A)

Line 66: Typo: replace "extension" with "extensions"

Line 66: by "large protruding extensions", do you mean to say "significant advances" or "large advances"? To me, "extension" reads more like a noun describing the geometry of the outlet glacier, rather than "advance", which reads like a verb describing the forward surge of the glacier front.

Fig. 1: I can't see any gradation in the red part of the colorbar in Fig. 1A -- it looks uniformly red.

Line 136-138: I like this sentence—it gives the reader good intuition for the extent of retreat.

Line 149: replace "which exhibited a thickening rate of +0.53 m/yr" with "which exhibited thickening at a rate of +0.53 m/yr".

Line 161: replace "regain half-way of its 1937..." with "regain half of its 1937..."

Line 195: add "pronounced" before regional?

Line 226: replace "show" with "suggest" (since it's a model reanalysis product, rather than true "data")?

Line 228: replace "it" with "snowfall"

Reviewer #2 (Remarks to the Author):

I was positive about the manuscript in the previous round of review and it has been improved since then. In summary I think the authors provide some very interesting, novel and robust observations. Any observations from the pre-satellite era are especially valuable and I am convinced by their argument that these trends in surface elevation are driven by change in snowfall. However, I still think the discussion gets a little confused in a small number of places, particularly when discussing processes surrounding glacier dynamics. After these relatively small issues have been addressed I would recommend the publication of this study.

Comments

Line 12: The new opening sentence in the abstract is a little awkward. Please re-write, in general I do not think stability is a good word to use in this context

Line 146-149: Where do these elevation change rates come from? No reference?

Line 194-212: Section: Controls of frontal variations and weakening of land-fast sea-ice:

Firstly, I think the heading needs to be edited, this section does not really discuss the controls on weakening landfast sea-ice. Secondly, I still find parts of this paragraph a little confusing, Line 203 states:

`Furthermore, the 2016-2018 retreat was coupled to a decrease in surface elevation and an

acceleration in flow, suggesting an important impact of sea ice on glacier dynamics'

But really the speed-up and decrease in surface elevation is only at *some* outlet glaciers in Lutzow-Holm Bay. Speed-up is only at Shirase, Skallen, Telen – not at Honnor or Langhovde (This one actually slowed down, see Fig 5 Kondo et al). Arguably, the link between speed-up and terminus position at Telen is also dubious because the speed-up predates terminus retreat by several years (Fig 5. Kondo et al). The alternative explanation here is that while landfast sea-ice is clearly important in determining terminus position, it plays no, or very limited role in ice speed and dynamics and instead melt rates near the grounding line are more important. This is because most of the ice shelves and ice tongues in LHB are highly fractured to the point where it is debatable whether the individual blocks of ice are in contact with each other and nearly all of terminus change is in the open ocean (i.e. no detachment from fjord walls or pinning points). This to me would suggest that the sections of the ice shelves that have fluctuated in this region are passive and offer very little buttressing and therefore have a limited impact on ice speed and dynamics. I appreciate that you more or less state this in later in the paragraph (Line 207), but the way the paragraph is constructed at the moment is a little confusing and contradictory. Thus, I would consider removing the sentence on Line 203, mentioned above, there is no evidence in this study or Kondo et al that landfast sea ice in LHB plays any role in grounded ice dynamics.

Line 238: I am not sure that the absence of dynamic mass loss supports increased precipitation. It is the combination of the ERA snowfall trends and your surface elevation change results that supports this hypothesis. Please consider removing this sentence. For example, flipping this around, precipitation has increased in parts of West Antarctica and the Peninsula, but glaciers are still dynamically losing mass.

Line 254: Could you not just compare like for like? Altimetry trends at the same location as you have extracted trends from the DEMs?

Reviewer #3 (Remarks to the Author):

Using historical airborne remote sensing data from 21 glaciers, Domgaard and others present a unique record of East Antarctic outlet glacier dynamics. The presented analysis spans 85 years and analyses calving front position changes of 21 glaciers, elevation changes of 12 glaciers and flow velocity of four glaciers. These measurements indicate a stable to slightly positive mass balance for the investigated basins. Even though, the number of collected measurements is very limited, the dataset is a valuable addition as remote sensing data over East Antarctica is very rare before the era of earth observation satellites. Hence, the effort the authors have put into data collection and processing should be appreciated.

The manuscript has already undergone several review cycles and improved based on comments by four previous reviewers. Therefore, most of the major concerns have already been addressed. That's why I will only add a few comments on points that haven't been discussed yet or weren't addressed fully in the revised version:

With the title "Early aerial expedition photos reveal 85 years of glacier growth and stability in East Antarctica" I would expect a larger study area spanning a wide area of East Antarctica also including basins recently losing mass such as Wilkes Land. Hence, for me the title suggests an entire stable East Antarctic ice sheet where glacier growth prevails for 85 years. But the small study area presented here cannot prove this. As the title has already been changed several times and Reviewer 1 within the first review round already raised this issue, I leave it up to the editor and authors whether they change the title again. One option could be: "Early aerial expedition photos reveal 85 years of glacier growth and stability in parts of East Antarctica".

After several review rounds it seems that the manuscript is no longer consistent on the conclusion that observed ice thickness changes can be linked to snowfall trends. In the abstract you say (L25): "... and link observed ice thickness changes to regional long-term trends in snowfall since 1940". Whereas in the conclusion you state (L267): "...we are unable to further deduct specific drivers of observed changes". Please make sure that your statement is consistent throughout the manuscript.

In line 217ff you mention that all regions rarely experienced above zero-degree temperatures. First, for me it is not clear how you calculated the summer temperatures in Fig. 4. Did you calculate an average temperature for all three months per year? In regard to surface melt it would be very important to consider extremes in temperature as well. E.g. for supraglacial lake formation it has been shown that positive degree days matter (see Dirscherl et al. 2021). Especially for the station data it seems that there were seasons with temperatures above zero degrees that should be investigated.

In the supplementary, I searched for figures showing the time series of glacier termini dynamics but couldn't find them. I think it would be a valuable addition to provide a figure with all delineated frontal positions to see what changes where measured by your centerline approach. Measuring calving front change can be prone to errors as calving front delineation is a very subjective task and the method applied for measuring terminus change matters (Lea et al. 2014).

Specific comments:

L115: Do you mean vertical uncertainty here?

L177: This sentence reads a bit misleading to me. Maybe better: In total, all glaciers along Christensen Coast experienced an absolute elevation increase....

L187: Can you explain why there are so high uncertainties during 2006 and 2013?

L243: twice "records"

L362: What is the vertical uncertainty of the REMA DEM and did you account for this when calculating errors for surface elevation changes?

References

M. C. Dirscherl, A. J. Dietz, und C. Kuenzer, „Seasonal evolution of Antarctic supraglacial lakes in 2015–2021 and links to environmental controls“, *The Cryosphere*, Bd. 15, Nr. 11, Art. Nr. 11, Nov. 2021, doi: 10.5194/tc-15-5205-2021.

J. M. Lea, D. W. F. Mair, und B. R. Rea, „Evaluation of existing and new methods of tracking glacier terminus change“, *Journal of Glaciology*, Bd. 60, Nr. 220, Art. Nr. 220, 2014.

Overview

We would like to thank all three reviewers for their positive feedback and their recommendations for publications. We are pleased that they have recognized the significant improvements made to the manuscript since its initial submission.

We have carefully addressed all the general concerns and specific comments raised by the reviewers and have revised the manuscript accordingly. A detailed breakdown of the edits can be found below, with referee comments highlighted in blue and our responses provided in red.

Moreover, we have shortened the abstract, and edited figures, tables etc. to follow the editorial guidelines, as well as organized supplementary materials into a separate document. Furthermore, during the preparation of data files for the online repository, we identified a minor error in the calculation of uncertainties. This issue resulted in negligible adjustments to uncertainty estimates, at the centimeter scale, and has absolutely no implications on the strength of our findings or conclusions. We have updated the figures and manuscript according to these new estimates.

Reviewers Comments:

Reviewer #1 (Remarks to the Author):

In this manuscript, Domgaard et al. analyze a set of historical aerial photographs (dating back to 1936) of East Antarctica and apply structure-from-motion and image correlation techniques in order to quantify historical elevation change and glacier surface velocities. In the discussion, Domgaard et al. compare their reconstructions of surface elevation change to weather reanalysis data from East Antarctica to suggest that the contrasting positive vs. negative mass balance behavior of their investigated sub-regions can be explained by regional trends in snowfall. I found this manuscript to be clear and well-written. I reviewed two previous iterations of this paper, and I found this most recent version to be the most thorough, thoughtful, and well-balanced (as much as one can expect for a short-format paper!). I have no remaining major reservations and recommend publication. See below for minor comments and line edits.

General comments:

I found the description of the data and methodology to be significantly improved from previous versions! I also found the format of a "Data and Methods" section at the end of the main text to be well-suited to this paper. This format allows for the main text to be relatively streamlined (not bogged down by the methodology), but also prevents relegating the methods to the Supplementary Information, which I think would do a disservice to the research because the curation and analysis of these historical images is the primary contribution of the paper.

There may be an error in lines 320-321, where the authors say "...we align the images...at Medium accuracy setting, meaning that photos are downscaled to half of its original resolution". At least in the older versions of Agisoft Photoscan that I'm familiar with, the setting "Very high" (or "Ultra") corresponds to no downscaling, "High" corresponds to downscaling by a factor of 2, and "Medium" corresponds to downscaling by a factor of 4. That's a factor of 4 in each dimension, so in total the

image will be 1/16th of its original size. Is it the same for the latest model of Agisoft Metashape? That could be something to double-check before publishing...

Thank you for pointing this out. The Agisoft Metashape pro manual for version 1.7 (the version used for the image processing) states that medium setting causes image downscaling by a factor of 4, 2 times in each dimension. This has now been corrected in the manuscript.

Line comments (minor):

Line 13: I might suggest a slight modification of your current phrasing: "In order to determine if recent trends are independent of natural variability..." Perhaps it is more precise to say: "In order to determine if recent trends exceed the scale of natural variability"?

Corrected

Line 21: add "in this region" or "in this sector". So, the full sentence will read: "The aerial images reveal that the current absence of mass loss in this sector can be extended back 85 years."

Corrected

Lines 25-26: It might be easier to read if you split this phrase ("...and link observed ice thickness changes to regional long-term trends in snowfall since 1940") into its own sentence. Perhaps: "At Kemp and Mac Robertson and Ingrid Christensen Coast, we link observed ice thickness changes to regional long-term trends in snowfall since 1940."

The sentence has been changed to accommodate comments from you and reviewer 3, the sentence now reads: *"Along the coastline of Kemp and Mac Robertson, and Ingrid Christensen Coast, we observe a long-term moderate thickening of the glaciers since 1937 and 1960 with periodic thinning and decadal variability. In all regions, the long-term changes in ice thickness correspond with the trends in snowfall since 1940."*

Line 43: change "through" to "from"

Corrected

Line 50: replace "historical climate data" with "historical climatic and glaciological data"

This sentence and the provided reference only concerns climate-related research and not glaciological data.

Line 53: "exploit" is an aggressive term. Replace with "leverage"?

Replaced with 'utilize'

Line 57: The word "data" should be plural: replace "this data" with "these data" or "these images"

Corrected to 'these data'

Line 64: Modify (Fig. 1) to (Fig. 1A)

Corrected

Line 66: Typo: replace “extension” with “extensions”

Corrected

Line 66: by “large protruding extensions”, do you mean to say “significant advances” or “large advances”? To me, “extension” reads more like a noun describing the geometry of the outlet glacier, rather than “advance”, which reads like a verb describing the forward surge of the glacier front.

By "large protruding extensions," we refer to the presence of large ice tongues. We have edited the text to reflect this: *“Some have large floating ice tongues, while others have their frontal position close to the grounding line.”*

Fig. 1: I can’t see any gradation in the red part of the colorbar in Fig. 1A -- it looks uniformly red.

We have edited the figure 1A to make gradation in the colorbar clearer.

Line 136-138: I like this sentence—it gives the reader good intuition for the extent of retreat.

Thank you

Line 149: replace “which exhibited a thickening rate of +0.53 m/yr” with “which exhibited thickening at a rate of +0.53 m/yr”.

Corrected

Line 161: replace “regain half-way of its 1937...” with “regain half of its 1937...”

Corrected

Line 195: add “pronounced” before regional?

Corrected

Line 226: replace “show” with “suggest” (since it’s a model reanalysis product, rather than true “data”)?

Corrected

Line 228: replace “it” with “snowfall”

Corrected

Reviewer #2 (Remarks to the Author):

I was positive about the manuscript in the previous round of review and it has been improved since then. In summary I think the authors provide some very interesting, novel and robust observations. Any observations from the pre-satellite era are especially valuable and I am convinced by their argument that these trends in surface elevation are driven by change in snowfall. However, I still think the discussion gets a little confused in a small number of places, particularly when discussing processes surrounding glacier dynamics. After these relatively small issues have been addressed I

would recommend the publication of this study.

Comments

Line 12: The new opening sentence in the abstract is a little awkward. Please re-write, in general I do not think stability is a good word to use in this context

The sentence has been edited and now reads: *“During the last decades, several sectors in Antarctica have transitioned from glacial mass balance equilibrium to mass loss.”*

Line 146-149: Where do these elevation change rates come from? No reference?

They come from Smith, B. et al. Pervasive ice sheet mass loss reflects competing ocean and atmosphere processes. *Science* 368, 1239–1242 (2020). This is mentioned in the method section, but a reference has now been added to the main text.

Line 194-212: Section: Controls of frontal variations and weakening of land-fast sea-ice:

Firstly, I think the heading needs to be edited, this section does not really discuss the controls on weakening landfast sea-ice. Secondly, I still find parts of this paragraph a little confusing, Line 203 states:

We have edited the titled to better reflect the statements in the text. It now reads:

“Frontal variations and decadal weakening of land-fast sea-ice”

‘Furthermore, the 2016-2018 retreat was coupled to a decrease in surface elevation and an acceleration in flow, suggesting an important impact of sea ice on glacier dynamics’

But really the speed-up and decrease in surface elevation is only at **some** outlet glaciers in Lutzow-Holm Bay. Speed-up is only at Shirase, Skallen, Telen – not at Honnor or Langhovde (This one actually slowed down, see Fig 5 Kondo et al). Arguably, the link between speed-up and terminus position at Telen is also dubious because the speed-up predates terminus retreat by several years (Fig 5. Kondo et al). The alternative explanation here is that while landfast sea-ice is clearly important in determining terminus position, it plays no, or very limited role in ice speed and dynamics and instead melt rates near the grounding line are more important. This is because most of the ice shelves and ice tongues in LHB are highly fractured to the point where it is debatable whether the individual blocks of ice are in contact with each other and nearly all of terminus change is in the open ocean (i.e. no detachment from fjord walls or pinning points). This to me would suggest that the sections of the ice shelves that have fluctuated in this region are passive and offer very little buttressing and therefore have a limited impact on ice speed and dynamics. I appreciate that you more or less state this in later in the paragraph (Line 207), but the way the paragraph is constructed at the moment is a little confusing and contradictory. Thus, I would consider removing the sentence on Line 203, mentioned above, there is no evidence in this study or Kondo et al that landfast sea ice in LHB plays any role in grounded ice dynamics.

Thank you for pointing this out. We have edited the sentences concerning the landfast sea ice and its role in grounded ice dynamics. It now reads:

“Additionally, land-fast sea ice played an important role in the observed simultaneous frontal retreat of the glaciers in Lützow-Holm Bay in the 1980s and again from 2016 to 2018. Moreover, the 2016-2018 retreat was coupled to a decrease in surface elevation and an acceleration in flow at Shirase, Skallen and Telen Glacier, whereas Honnörbrygga and Langhovde Glacier were unaffected. Similarly, our findings of a long-term frontal retreat at Honnörbrygga and Langhovde Glacier since 1937 does not coincide with any changes in the surface elevation of these glaciers (Fig 2), suggesting that these floating ice tongues have provided limited buttressing on a decadal time-scale.”

Line 238: I am not sure that the absence of dynamic mass loss supports increased precipitation. It is the combination of the ERA snowfall trends and your surface elevation change results that supports this hypothesis. Please consider removing this sentence. For example, flipping this around, precipitation has increased in parts of West Antarctica and the Peninsula, but glaciers are still dynamically losing mass.

We are arguing that the resulting surface increase, is caused by the precipitation since there is no dynamic change. Not that the dynamic change is not occurring because of the increased precipitation. This is now more clear in the text:

“Nevertheless, we hypothesize that the observed increase in surface elevation since 1937 in Kemp and Mac Robertson Land (Fig. 4) is likely a result of changes in precipitation patterns. The absence of dynamic ice changes, observed from stable frontal positions since the 1930s and flow velocities since the 1950s (Fig. 2 and Fig. 3) suggest that the observed ice surface increase results from the increase in precipitation (Fig. 4).”

Line 254: Could you not just compare like for like? Altimetry trends at the same location as you have extracted trends from the DEMs?

During the yearly decades, the referenced altimetry data is captured using Geosat and ERS radar altimeters, with footprints spanning multiple kilometers and extremely high uncertainties in areas where surface slope exceeds just 0.6° (Nilsson et al. 2022) - making it specifically uncertain in coastal regions. During the era of ICESat and ICESat-2 acquisitions (2003-2021) when uncertainties are significantly reduced, we already conduct such a point-specific comparison (Fig. 2).

Reviewer #3 (Remarks to the Author):

Using historical airborne remote sensing data from 21 glaciers, Domgaard and others present a unique record of East Antarctic outlet glacier dynamics. The presented analysis spans 85 years and analyses calving front position changes of 21 glaciers, elevation changes of 12 glaciers and flow velocity of four glaciers. These measurements indicate a stable to slightly positive mass balance for the investigated basins. Even though, the number of collected measurements is very limited, the dataset is a valuable addition as remote sensing data over East Antarctica is very rare before the era of earth observation satellites. Hence, the effort the authors have put into data collection and processing should be appreciated.

The manuscript has already undergone several review cycles and improved based on comments by four previous reviewers. Therefore, most of the major concerns have already been addressed. That's why I will only add a few comments on points that haven't been discussed yet or weren't addressed fully in the revised version:

With the title “Early aerial expedition photos reveal 85 years of glacier growth and stability in East Antarctica” I would expect a larger study area spanning a wide area of East Antarctica also including basins recently losing mass such as Wilkes Land. Hence, for me the title suggests an entire stable East Antarctic ice sheet where glacier growth prevails for 85 years. But the small study area presented here cannot prove this. As the title has already been changed several times and Reviewer 1 within the first review round already raised this issue, I leave it up to the editor and authors whether they change the title again. One option could be: “Early aerial expedition photos reveal 85 years of glacier growth and stability in parts of East Antarctica”.

We prefer to keep the currently title, but will leave it to the editor to decide if it should be changed to “parts of East Antarctica. We believe the former is the best as no other parts have a comparable long-term record.

After several review rounds it seems that the manuscript is no longer consistent on the conclusion that observed ice thickness changes can be linked to snowfall trends. In the abstract you say (L25): “... and link observed ice thickness changes to regional long-term trends in snowfall since 1940”. Whereas in the conclusion you state (L267): “...we are unable to further deduct specific drivers of observed changes”. Please make sure that your statement is consistent throughout the manuscript.

We have adjusted the wording in the abstract to ensure better consistency throughout the manuscript. We have edited the abstract to reflect the statements in the discussion. It now reads: *“In all regions, the long-term changes in ice thickness correspond with the trends in snowfall since 1940”*

In line 217ff you mention that all regions rarely experienced above zero-degree temperatures. First, for me it is not clear how you calculated the summer temperatures in Fig. 4. Did you calculate an average temperature for all three months per year? In regard to surface melt it would be very important to consider extremes in temperature as well. E.g. for supraglacial lake formation it has been shown that positive degree days matter (see Dirscherl et al. 2021). Especially for the station data it seems that there were seasons with temperatures above zero degrees that should be investigated.

We calculate the mean austral summer temperature as the average of the December value for year n and the January and February values of year $n+1$. As you correctly state, we do observe a few years with mean summer temperature above zero at Davis Station. We have now added a sentence on this in the manuscript. It reads: *“However, at Davis station we do observe periods of above zero-degree temperature during 1970s and 2000s, which have caused intervals of increased ablation in this region near sea level.”*

In the supplementary, I searched for figures showing the time series of glacier termini dynamics but couldn't find them. I think it would be a valuable addition to provide a figure with all delineated frontal positions to see what changes where measured by your centerline approach. Measuring calving front change can be prone to errors as calving front delineation is a very subjective task and the method applied for measuring terminus change matters (Lea et al. 2014).

We have now included figures (Fig. S25, S26 and S27) in the supplementary material illustrating the time series of glacier termini fluctuations.

Specific comments:

L115: Do you mean vertical uncertainty here?

Yes. The text now reads: *“elevation uncertainty”*

L177: This sentence reads a bit misleading to me. Maybe better: In total, all glaciers along Christensen Coast experienced an absolute elevation increase....

Corrected

L187: Can you explain why there are so high uncertainties during 2006 and 2013?

The uncertainties are related to the quality of the imagery used for velocity tracking. From 2006 to 2013, velocities are derived from images captured by Landsat 4, 5, and primarily Landsat 7 (with scan-line error). Landsat 8 was launched in 2013. Additionally, we have now added more recent velocity observations to the figure.

L243: twice “records”

Corrected

L362: What is the vertical uncertainty of the REMA DEM and did you account for this when calculating errors for surface elevation changes?

The REMA DEM exhibits absolute uncertainties of less than 1m, with relative uncertainties at the decimeter level (Howat et al., 2019). As all of our estimates are calculated relative to the REMA DEM, we are only concerned with the relative accuracy. We adopt a conservative estimate of 1m for all of our surface elevation change calculations. This is now more clear in the text:

“As a result, the number of circles placed varies between the glaciers. All elevation changes are calculated relative to the REMA reference DEM strip, as differencing two historical DEMs with data gaps results in more areas with no data. We incorporate a conservative uncertainty estimate of 1m for the REMA DEM (Howat et al., 2019) into our calculations. “

References

M. C. Dirscherl, A. J. Dietz, und C. Kuenzer, „Seasonal evolution of Antarctic supraglacial lakes in 2015–2021 and links to environmental controls“, *The Cryosphere*, Bd. 15, Nr. 11, Art. Nr. 11, Nov. 2021, doi: 10.5194/tc-15-5205-2021.

J. M. Lea, D. W. F. Mair, und B. R. Rea, „Evaluation of existing and new methods of tracking glacier terminus change“, *Journal of Glaciology*, Bd. 60, Nr. 220, Art. Nr. 220, 2014.

References

Howat, I. M., Porter, C., Smith, B. E., Noh, M.-J., and Morin, P.: The Reference Elevation Model of Antarctica, *The Cryosphere*, 13, 665–674, <https://doi.org/10.5194/tc-13-665-2019>, 2019.

Smith, B. et al. Pervasive ice sheet mass loss reflects competing ocean and atmosphere processes. *Science* 368, 1239–1242 (2020).

Nilsson, J., Gardner, A. S., and Paolo, F. S.: Elevation change of the Antarctic Ice Sheet: 1985 to 2020, *Earth Syst. Sci. Data*, 14, 3573–3598, <https://doi.org/10.5194/essd-14-3573-2022>, 2022.

REVIEWERS' COMMENTS

Reviewer #2 (Remarks to the Author):

I thank the authors for responding to the few minor points raised in the previous review round. After the most recent round of edits, I am now satisfied the manuscript is ready for publication and I have no further comments. A really interesting study!

Reviewer #3 (Remarks to the Author):

I am happy to see that the authors revised all raised points and answered all reviewer questions in detail. I no longer have any further comments and consider the manuscript worthy of publication.